# The Second Generation Antibody-Drug Conjugate SYD985 Overcomes Resistances to T-DM1

**DOI:** 10.3390/cancers12030670

**Published:** 2020-03-13

**Authors:** Mercedes Nadal-Serrano, Beatriz Morancho, Santiago Escrivá-de-Romaní, Cristina Bernadó Morales, Antonio Luque, Marta Escorihuela, Martín Espinosa Bravo, Vicente Peg, Fred A. Dijcks, Wim H.A. Dokter, Javier Cortés, Cristina Saura, Joaquín Arribas

**Affiliations:** 1Preclinical and Translational Research Program, Vall d’Hebron Institute of Oncology (VHIO), 08035 Barcelona, Spain; mnadal@vhio.net (M.N.-S.); bmorancho@vhio.net (B.M.); cbernado@vhio.net (C.B.M.); aluque@vhio.net (A.L.); mescorihuela@vhio.net (M.E.); jacortes@vhio.net (J.C.); 2Breast Cancer and Melanoma Group, Vall d’Hebron Institute of Oncology (VHIO), 08035 Barcelona, Spain; sescriva@vhio.net (S.E.-d.-R.); maespino@vhebron.net (M.E.B.); vpeg@vhebron.net (V.P.); csaura@vhio.net (C.S.); 3Medical Oncology Department, Vall d´Hebron University Hospital (HUVH), 08035 Barcelona, Spain; 4Preclinical R&D, Synthon Biopharmaceuticals BV, 6545 CM Nijmegen, The Netherlands; Fred.Dijcks@synthon.com (F.A.D.);; 5Centro de Investigación Biomédica en Red de Cáncer (CIBERONC), 08035 Barcelona, Spain; 6Department of Biochemistry and Molecular Biology, Universitat Autónoma de Barcelona, Campus de la UAB, 08193 Bellaterra, Spain; 7Institució Catalana de Recerca i Estudis Avançats, 08010 Barcelona, Spain

**Keywords:** breast cancer, HER2, antibody-drug conjugate, SYD985, T-DM1, patient-derived xenograft

## Abstract

Trastuzumab-emtansine (T-DM1) is an antibody-drug conjugate (ADC) approved for the treatment of HER2 (human epidermal growth factor receptor 2)-positive breast cancer. T-DM1 consists of trastuzumab covalently linked to the cytotoxic maytansinoid DM1 via a non-cleavable linker. Despite its efficacy, primary or acquired resistance frequently develops, particularly in advanced stages of the disease. Second generation ADCs targeting HER2 are meant to supersede T-DM1 by using a cleavable linker and a more potent payload with a different mechanism of action. To determine the effect of one of these novel ADCs, SYD985, on tumors resistant to T-DM1, we developed several patient-derived models of resistance to T-DM1. Characterization of these models showed that previously described mechanisms—HER2 downmodulation, impairment of lysosomal function and upregulation of drug efflux pumps—account for the resistances observed, arguing that mechanisms of resistance to T-DM1 are limited, and most of them have already been described. Importantly, SYD985 was effective in these models, showing that the resistance to first generation ADCs can be overcome with an improved design.

## 1. Introduction

The receptor tyrosine kinase (RTK) HER2 is overexpressed in different tumors, including approximately 25% of breast cancers [1]. Its tumor-driving activity and presence at the cell surface has made HER2 a prototypical target of precision therapeutic agents. These include the monoclonal antibodies trastuzumab and pertuzumab, which are currently the first line of treatment for HER2-positive breast cancers, and the antibody-drug conjugate (ADC) T-DM1, as the second line [2,3].

T-DM1 comprises trastuzumab linked to the maytansinoid DM1 via a stable thioether linker [4]. Binding to HER2 triggers the endocytosis of T-DM1 and its subsequent delivery to the lysosomes, where DM1 is released and transported to the cytosol. There, DM1 binds to tubulin and inhibits microtubule assembly causing mitotic catastrophe and apoptosis (for a recent comprehensive review of the mechanism of action of T-DM1, see [5]). Despite its clinical efficacy, breast cancers frequently become resistant to T-DM1 [3,6].

Using established breast cancer cell lines, several groups have identified mechanisms of resistance to T-DM1. These include downregulation of HER2 [7,8,9], activation of alternative RTKs or intracellular signaling pathways downstream of HER2 [7], upregulation of Bcl-2/X_L_ [8], defective intracellular routing [9] defective lysosomal function [10,11,12] and upregulation of multidrug resistance transporters [13,14,15].

To overcome this resistance, two approaches have been proposed. One consists in combination therapies that include drugs to counteract resistance, such as T-DM1 plus inhibitors of cyclin dependent kinase 4 and 6 (cdk4/6) [16]. Since the cdk4/6 is a downstream effector of RTKs and intracellular signaling pathways downstream of HER2, its inhibition may counteract these resistances. The second approach is based on second generation ADCs targeting HER2. By modifying the linker to facilitate the release in the tumor of a more potent, membrane permeable, payload with a different mechanism of action (MoA), several mechanisms of resistance could be overcome.

SYD985 is a second generation ADC consisting of trastuzumab bound to a potent duocarmycin payload via a cleavable linker (vc-*seco*-DUBA) [17,18]. The membrane permeable duocarmycin alkylates DNA, resulting in DNA damage, mitochondrial stress, impaired DNA transcription and apoptosis. Due to its membrane permeable nature, duocarmycin also reaches neighboring cells, a process known as bystander killing effect [19]. SYD985, and other second generation ADCs targeting HER2, are being tested in the clinic with promising results [20,21].

In the current study we focused on patient-derived xenografts (PDXs) to determine the effect of SYD985 in models that are resistant to T-DM1. These models recapitulate the inter- and intratumoral heterogeneity of human tumors [22]. We found that different PDXs showed different sensitivities to T-DM1. Using PDXs sensitive to T-DM1, we generated several models of resistance in vitro and in vivo. Using the in vitro generated models, we characterized the mechanisms of resistance. Finally, we showed that, independently of the mechanism of resistance, SYD985 displays a remarkable activity on all models resistant to T-DM1. This result strongly suggests that this second-generation ADC is an efficient way to overcome resistance to T-DM1.

## 2. Results

### 2.1. Models of Resistance to T-DM1

We used a panel of four HER2-positive patient-derived xenografts (PDXs 118, 510, 515 and 580), and one triple negative PDX (549) with low but detectable levels of HER2 (Figure 1A–C). Three of the HER2-positive PDXs (118, 510 and 515) were sensitive to T-DM1. PDX580 was established from a tumor that progressed on T-DM1 and, as xenograft, maintained resistance to the treatment. As expected, the HER2 1 + PDX549 was also resistant (Figure 1D).

To generate models of acquired resistance, we used in vitro and in vivo approaches (Figure 2A). The in vitro models allow molecular characterization of the mechanisms of resistance while, arguably, in vivo models are closer to real tumors. Cells derived from PDX118 were chronically treated with T-DM1, in vitro, as described in [23] (Figure 2A). As a result, we obtained different resistant cells (R44, R55 and R200) that, compared to parental cells, showed significantly higher IC50s for T-DM1 (Figure 2B).

To obtain in vivo models, we chronically treated PDX118 and PDX510 with T-DM1. The growth of the resulting models (118vo-R and 510vo-R) was not prevented by treatment with T-DM1 (Figure 2C). In sum, we identified two HER2-positive PDXs resistant to T-DM1 and generated several models of acquired resistance by chronic in vitro or in vivo treatment.

### 2.2. Characterization of in vitro Resistant Models

We have previously partially characterized two of the resistant models (R44 and R55) and found a defect in lysosomal function that results in a higher lysosomal pH that may partially explain their resistance to T-DM1 [11]. To further support that compromising the lysosomal function affects the efficacy of T-DM1, we treated parental cells with the lysosomal inhibitors chloroquine and bafilomycin [24,25]. We used concentrations that did not affect cell proliferation, but increased lysosomal pH, as shown by the pH-sensitive fluorochrome pHrodo (Figure 3A). Both lysosomal inhibitors reduced the efficacy of T-DM1 (Figure 3B). Thus, an increase in lysosomal pH, similar to that previously shown in R44 and R55 cells [11], partially explains the resistance of these cells.

Quantification of the levels of HER2 (mRNA and protein) showed a ~ 50% downmodulation in R44 and R55 cells (Figure 3C). To determine if the downmodulation of HER2 contributed to the resistance to T-DM1, we transfected resistant cells with a vector encoding HER2 and selected resistant cells with levels of expression comparable to those of parental cells (Figure 3D). Overexpression of HER2 led to partial re-sensitization of resistant cells to T-DM1, showing that reduction in the levels of HER2 contributes to resistance to T-DM1 (Figure 3E). On the other hand, overexpression of HER2 in parental PDX118 cells did not decrease the IC50 (Figure 3D,E), indicating that the levels of HER2 expressed by the parental cells are sufficient to trigger a maximal response to T-DM1. These experiments show that downmodulation of HER2 contributes to the resistance of R44 and R55 cells. Thus, both impairment of lysosomal function [11] and downmodulation of HER2 (Figure 3C) account for the resistance to T-DM1 of these cells.

Since we did not detect any alteration in the lysosomal pH of R200 cells and their levels of HER2 are unaltered (Figure 3C), we explored additional mechanisms of resistance in these cells. We observed that, compared to parental cells, R200 cells were resistant to free active maytansinoid (S-Me-DM1) (Figure 3F), indicating that drug efflux contributes to resistance in these cells. In fact, elevated expression of ATP-binding cassette (ABC) transporters such as ABCC1 (also known as MRP1) reduce the effectivity of T-DM1 by increasing the efflux of the payload DM1 from cancer cells [26]. Consistently, treatment with Reversan, a selective inhibitor of ABCC1 [27], had no effect on the sensitivity of PDX118, R44 or R55 cells but significantly increased the sensitivity of R200 to T-DM1 (Figure 3G), further supporting that the drug efflux pump contributes to the resistance of R200 cells. In summary, the characterization of different in vitro models of acquired resistance to T-DM1 identified several previously described mechanisms: HER2 downmodulation, impairment of lysosomal function and upregulation of drug-efflux pumps.

### 2.3. Response of T-DM1-Resistant Models to SYD985

All the in vitro-generated T-DM1-resistant models showed sensitivity to SYD985 comparable to that of parental cells (Figure 4A). This result shows that decreased HER2 levels, impairment of lysosomal function or increased drug efflux do not affect the efficacy of SYD985. Confirming the ability of SYD985 to overcome lysosomal impairment, inhibition of lysosomal function with chloroquine had no effect on the cytotoxicity mediated by SYD985 (Figure 4B). Moreover, increasing the levels of HER2 in R55 or R44 cells (Figure 4C) or inhibition of drug efflux by Reversan in R200 cells (Figure 4D) did not affect SYD985 cytotoxicity. In sum, the effect of SYD985 is unaffected by different mechanisms that impair the effect of T-DM1.

We subsequently evaluated the efficacy of SYD985 on tumors resistant to T-DM1. The two models of in vivo acquired resistance showed markedly reduced levels of HER2 (Figure 5A–D). The mechanism of HER2 downmodulation in the 118-voR is likely post-transcriptional since the levels of HER2 transcript and HER2 gene copy number remained largely unaffected (Figure 5A,D). In contrast, the levels of HER2 mRNA in the 510-voR cells were significantly lower than those of parental cells (Figure 5A), as was the HER2 gene copy number (Figure 5D), showing that the mechanism of downmodulation in these cells is due to loss in HER2 gene amplification. As expected, SYD985 had marked anti-tumoral effect on parental PDX118 and PDX510 (Figure 5E). Although not as efficacious as in the original PDXs, SYD985 had a significant anti-tumor effect on the two models of acquired resistance, 118-voR and 510-voR (Figure 5E). This result confirms the in vitro analyses showing that downmodulation of HER2 leads to resistance to T-DM1, but not to resistance to SYD985.

Next, we tested SYD985 on a model primarily resistant to T-DM1, PDX580. Although the mechanism of resistance of this PDX is uncharacterized, SYD985 effectively reduced tumor growth further supporting the efficacy of this second-generation ADC on tumors resistant to T-DM1 (Figure 5F). Finally, the model of triple negative PDX549, which expresses low but detectable levels of HER2 (Figure 1A–C) and is resistant to T-DM1 (Figure 1D), is also sensitive to SYD985 (Figure 5F). In summary, different mechanisms of resistance to T-DM1 can be overcome with an ADC directed against the same target, HER2, but with a cleavable linker and a different payload.

## 3. Discussion

As is the case with any efficacious anti-cancer therapy, some target cells are unsensitive to T-DM1 and some, after an initial response, develop resistance. Using HER2-positive breast cancer cell lines chronically treated with T-DM1, multiple mechanisms of resistance have been identified (for reviews, see [5,13]). Notably, despite the variety of mechanisms described, downmodulation of HER2 has been repeatedly identified by different groups [15,25,27].

In the study presented here, we used patient-derived xenografts, which are considered an experimental model that resembles clinically observed inter- and intratumoral heterogeneity [22]. Of the T-DM1 resistant models characterized in our study, two out of three developed in vitro and two out of two developed in vivo, showed downmodulation of HER2, further supporting that this may be the most frequent mechanism of resistance.

In agreement with this conclusion, in the clinical setting analysis of tumors from patients treated with neoadjuvant trastuzumab that did not achieve complete pathological response showed that 30% displayed reduced HER2 expression through the loss of gene amplification [28]. It should be kept in mind that this observation may represent an underestimation of the frequency of HER2 downmodulation in resistant tumors, as our results show that HER2 can be downmodulated not only by the loss of gene amplification but also through post-transcriptional mechanisms in cells that have not lost gene amplification.

Additional mechanisms of resistance already described include defects in intracellular degradation of T-DM1 and resistance to the cytotoxic effect of the maytansinoid payload through drug efflux. We observed these mechanisms in some of the in vitro models analyzed here and in a previous study [11]. Thus, using patient-derived xenografts as models, we identified only mechanisms previously described, indicating that the number of mechanisms of resistance is limited and that most of them have been already found.

According to our results obtained with an in vitro model, SYD985 overcomes defective lysosomal function. This result is in agreement with previous findings showing that SYD985 does not require the acidity of the lysosomal compartment of wild type cells and that the payload can be released at higher pHs [19]. Additionally, we showed that SYD985 is effective in resistant cells that can extrude DM1, indicating that a drug-efflux pump inhibited by Reversan does not affect the cytotoxic effect of duocarmycin. Finally, SYD985 was also effective against a PDX primarily resistant to T-DM1, which expressed high levels of HER2 and whose mechanism of resistance is unknown.

SYD985 is remarkably effective in the various models that developed resistance through downmodulation of HER2 in vitro and in vivo. This result supports that SYD985 may be effective against tumors expressing relatively low levels of HER2, in agreement with clinical data [29]. We confirmed this view by showing the anti-tumor effect on a triple negative PDX expressing detectable levels of HER2. In summary, our results warrant further clinical development of second-generation ADCs, such as SYD985, as we predict that they will be effective on tumors with different types of resistances to T-DM1.

## 4. Material and Methods

### 4.1. Reagents

Trastuzumab and trastuzumab-emtansine (T-DM1) were obtained from a local pharmacy. SYD985 was obtained from Synthon Biopharmaceuticals BV (Nijmegen, the Netherlands). S-Me-DM1 was purchased from Abcam (Cambridge, UK), bafilomycin A1 (BafA1) and chloroquine from Sigma-Aldrich (St. Louis, MO, USA) and Reversan from Millipore (Burlington, MA, USA).

### 4.2. Tumor Samples

Human breast tumors used in this study were from biopsies or surgical resections at Vall d’Hebron University Hospital and were obtained following institutional guidelines. PDX515 and PDX549 were originated from a biopsy of the primary tumor and PDX118, PDX510 and PDX580 were generated from a biopsy of skin, a biopsy below the reconstruction and a lymph node biopsy respectively. The IRBs at Vall d’Hebron University Hospital provided approval for this study in accordance with the Declaration of Helsinki. Written informed consent was obtained from all patients who provided tissue. This research has been approved by Vall d’Hebron University Hospital ethic committee (Comité Ético de Investigación Clínica) on 1st April 2011 (ethic code: PR(AG)173/2011).

### 4.3. Patient-Derived Xenografts (PDXs)

Six to eight-week old NOD.CB17-Prkdcscid/J (NOD/SCID) mice were purchased from Janvier Labs. Mice were maintained and treated in accordance with protocols approved by the Ethical Committee for the Use of Experimental Animals at the Vall d’Hebron Institute of Oncology on 25th October 2018 (ethic code: CEA-OH/10303/1). Fragments of human breast tumors were implanted into the fourth fat pad of the mice, which were supplemented with 1 μM 17ß-estradiol (Sigma-Aldrich) in the drinking water. One to two mice were used per sample to implant the patient biopsies, depending on its size.

### 4.4. In Vivo Treatments

Six to eight-week old NOD.CB17-Prkdcscid/J (NOD/SCID) mice carrying PDXs were treated intravenously with SYD985 (10 mg/kg), T-DM1 (15 mg/kg) or vehicle (PBS) every 3 weeks. Tumor measurements were recorded twice weekly using the formula: (length × width2) × (pi/6). Mice were sacrificed when the tumor volume approached 1500–2000 mm^3^. Tumor samples were then collected and immediately processed for later analysis.

### 4.5. Primary Cultures

For the establishment of cell cultures derived from PDXs, tumors were excised and cut into the smallest pieces possible with scalpel and digested in collagenase (300 U/mL) plus hyaluronidase (100 U/mL) (Sigma-Aldrich) in Dulbecco’s modified Eagle’s medium/F12 (DMEM/F12) media. After 1 h of incubation at 37 °C with shaking at 80 rpm, the mixture was filtered thought a 100 μm cell strainer. Red blood cell lysis was performed in digested tumors. After a wash with PBS, the cells from tumor were resuspended in DMEM/F12 supplemented with 10% FBS and 4 mmol/L L-glutamine (all from Gibco, Waltham, MA, USA).

### 4.6. Generation of T-DM1 Resistant Models

To generate models of acquired resistance, cells were chronically treated with T-DM1, in vitro or in vivo. For in vitro models, cells obtained from PDX118 were cultured in the presence of increasing concentrations of T-DM1 for 45–90 days until three resistant cell cultures were established. Cell cultures were tested for resistance in vitro following a dose response study of T-DM1 (from 0.03 to 2 nM) over 6 days. To establish T-DM1-resistant in vivo models, as previously described [23], mice carrying PDX118 or PDX510 were continuously treated with 15 mg/kg T-DM1 every 3 weeks. After 70–160 days, the tumors relapsed and their resistance to T-DM1 was confirmed after re-implantation into new mice.

### 4.7. Plasmids and shRNA Lentiviruses

Stable overexpressing HER2 cells were generated by lentiviral transduction of pRedZeo-HER2 vector, generated by a modification of the pRedZeo-CMV (SR10046PA-1, System Biosciences, Palo Alto, CA, USA) to allow HER2 overexpression.

### 4.8. Immunohistochemical Analysis

Patient-derived xenografts sample tissues were fixed in 4% Formaldehyde buffered to pH 7.0 (stabilized with methanol) for 24 h and then paraffin-embedded (FFPE). Tissue sections of 4 µm thickness were mounted on positively charged glass slides and immunostained with anti-HER2 (c-erbB-2) (CB11, #MU134-UCE, BioGenex, Fremont, CA, USA). The whole blot can be found at Appendix A.

### 4.9. T-DM1 Labeling and Analysis of Lysosomal Activity 

T-DM1 was labeled with pHrodo Red succinimidyl ester (pHrodo Red, Life Technologies, Carlsbad, CA, USA), a non-fluorescent dye at neutral pH that fluoresces brightly in acidic environments, according to the manufacturer’s instructions. Cells were treated with 1 nM bafilomycin A1 or 1 μM chloroquine and 100 ng/mL pHrodo-labeled T-DM1. Then, labeled-cells were collected and analyzed by flow cytometry on LSR Fortessa (BD Biosciences, Franklin Lakes, NJ, USA). 

### 4.10. mRNA Expression

RNA isolation from 20–30 mg fresh-frozen PDXs samples or at least 5 × 10^5^ cultured cells was performed as recommended by the supplier (RNeasy Mini Kit, Qiagen, Hilden, Germany). One microgram of the total RNA was reverse-transcribed to cDNA by using the High Capacity cDNA Reverse Transcription Kit (Applied Biosystems, Foster City, CA, USA). Quantitative reverse transcription-polymerase chain reaction (qRT-PCR) was performed with Taqman primers (Applied Biosystems) for HER2 (HS01001580_m1) and GAPDH (Hs02758991_g1) using an ABI 7900HT Sequence Detection System (city, if any state abbreviation, country). Each sample was assayed in triplicate and normalized by using GAPDH gene expression as a reference gene. The Ct (Cycle Threshold) values of the real-time PCR were extracted from each assay with SDSv2.0 software tool (Applied Biosystems) and analyzed using DataAssist Software (Applied Biosystems) following the manufacturer’s instructions. 

### 4.11. Cell Proliferation Assay

Cells (4 × 10^3^ per well) were seeded in 96-well plates and incubated for 72 h. Then, cells were treated with T-DM1 or SYD985 at the described concentrations. At the indicated time points, cells were fixed by replacing the growth medium with 10% glutaraldehyde for 15 min at room temperature. After three washes with water, 0.1% crystal violet solution was added to each well and incubated 30 min, then plates were washed and air-dried. Optical density (OD) was measured at 560 nm after resuspension in 10% acetic acid.

### 4.12. Protein Extraction and Immunoblotting

For immunoblotting, 20–30 mg of fresh-frozen PDXs samples or at least 1 × 10^6^ cultured cells were lysed in NP-40 buffer containing 50 mM Tris-HCl, pH 7.4; 150 mM NaCl, 2 mM EDTA, pH 8.0; 10% (v/v) glycerol and 1% (v/v) Nonidet-P40 plus protease and phosphatase inhibitors. Cell lysates were sonicated once at a frequency of 20 kHz for 10 seconds. Then samples were centrifuged at 14,000· g for 20 min at 4°C and protein content was measured using the Bio-Rad DC Protein Assay (Bio-Rad, Hercules, CA, USA). Equal amounts of protein were resolved by 8–15% SDS-PAGE gel and electrotransferred onto nitrocellulose membranes. Proteins were visualized by Western blotting using antibodies directed against the indicated antigens: HER2 (CB11, #MU134-UCE, BioGenex) and human glyceraldehyde-3-phosphate dehydrogenase (GAPDH) (#Ab128915, Abcam, Cambridge, UK).

### 4.13. Copy Number Analysis

Genomic DNA was obtained from frozen tumor samples using the QIAmp DNA Mini kit following manufacturer’s protocol (Qiagen). Ten ng of DNA per sample were run in quadruplicate in an Applied Biosystems 7900HT using ErbB2 TaqMan Copy Number Assays (Hs00641606_cn) and RNase P TaqMan Copy Number Reference Assay (4403326). Copy number was calculated with the Applied Biosystems CopyCaller Software.

### 4.14. Statistical Analyses

Data are presented as average ± standard deviation and were analyzed by the two-sided Student’s *t*-test. Results were considered to be statistically significant at a P value of less than 0.05. All statistical analyses were conducted by using the GraphPad Prism version 6.0 software (GraphPad Software, San Diego, CA, USA).

## 5. Conclusions

Our results indicate that SYD985 will be effective on tumors with different types of resistances to T-DM1. In our patient-derived models, T-DM1 resistance is caused by several mechanisms: HER2 downmodulation, impairment of lysosomal function and upregulation of drug efflux pumps. Importantly, SYD985 overcomes all these resistances and has a potent anti-tumoral effect.

## Figures and Tables

**Figure 1 cancers-12-00670-f001:**
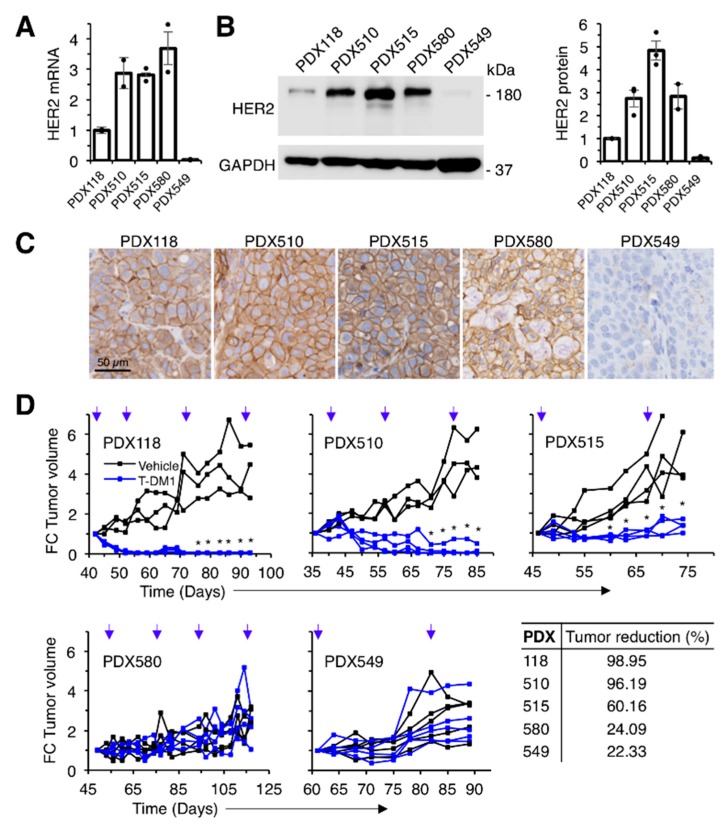
Effect of T-DM1 on the growth of breast cancer PDXs with different HER2 expression levels. (**A)** Relative HER2 mRNA expression was determined by RT-PCR in a collection of breast cancer PDXs. Data were normalized to GAPDH mRNA. Averages and standard deviation of two independent determinations are represented; (**B**) Lysates from the same PDXs were analyzed by Western blot with the indicated antibodies (left panel) and densitometrical quantification was performed (right panel). Quantitative results are expressed as averages ± standard deviations of three independent experiments; (**C**) Immunohistochemical analysis of the expression of HER2 in the same PDXs; (**D**) NOD/SCID mice carrying the indicated PDXs were treated with vehicle or T-DM1 as indicated (blue arrows). Tumor volume was determined and expressed as fold-change relative to the initiation of treatment. Tumor reduction was calculated at the end of the experiment as the percentage of the average tumor volumes of the tumors treated with T-DM1 relative to tumors treated with vehicle. * *p* < 0.05, two-tailed Student *t* test.

**Figure 2 cancers-12-00670-f002:**
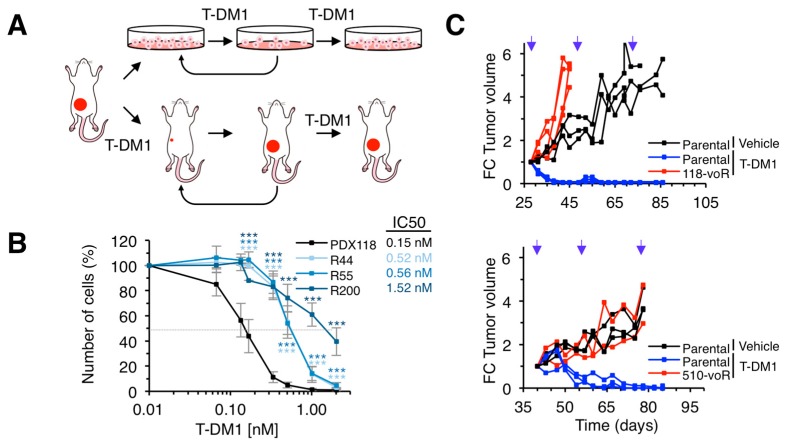
Generation of in vitro and in vivo models of acquired resistance to T-DM1. (**A**) Schematic drawing illustrating the strategies to generate in vitro and in vivo resistant models from PDXs. (**B**) The indicated cell cultures were treated with increasing concentrations of T-DM1 for 6 days. Cell numbers were estimated by crystal violet staining. Dashed grey line indicates 50% survival and IC50 is shown next to figure legend. Averages and standard deviation of six independent experiments are represented. ** *p* < 0.01, *** *p* < 0.001, two-tailed Student *t* test. (**C**) NOD/SCID mice carrying the indicated parental PDXs or the resistant PDXs obtained in vivo as shown in A were treated and analyzed as described in Figure 1D.

**Figure 3 cancers-12-00670-f003:**
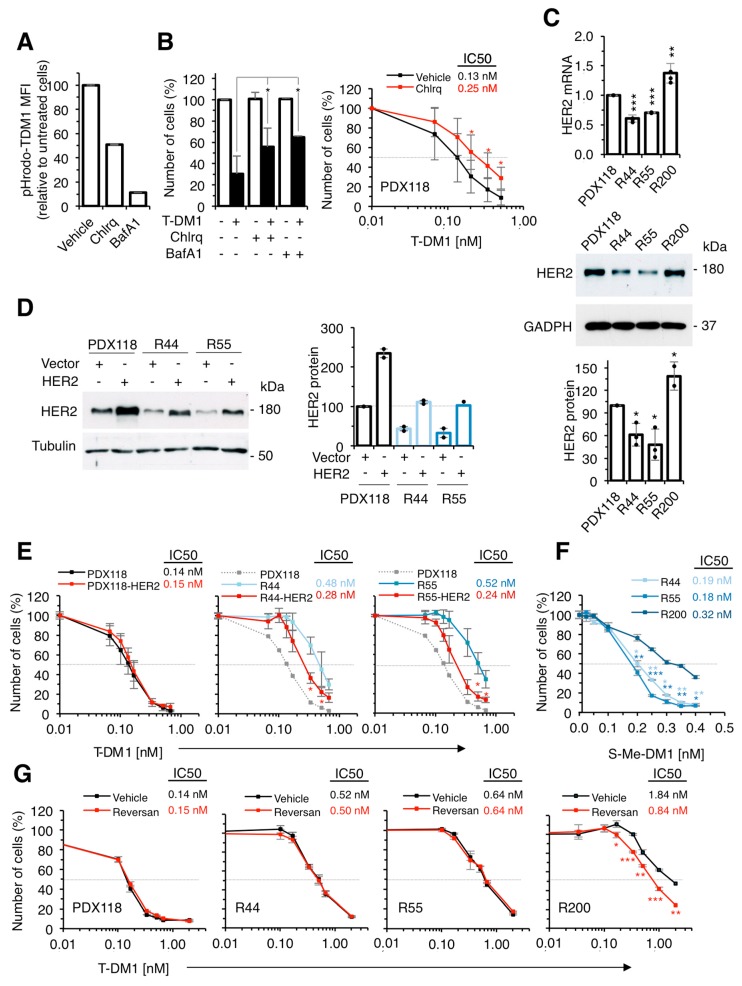
Characterization of in vitro resistant cells. (**A**) Cultures from PDX118 treated with vehicle, Chloroquine (Chlrq) or Bafilomycin A (BafA1) were stained with pHrodo-T-DM1 for 24 hours at 37 °C and analyzed by FACS. Results are expressed as a percentage of red-positive cells; (**B**) Left, Cell cultures derived from PDX118 treated with vehicle (-), Chloroquine or Bafilomycin A and T-DM1 as indicated were quantified by crystal violet staining and expressed as percentages relative to cells treated with vehicle. Results are averages and standard deviation of four independent experiments. * *p* < 0.05, two-tailed Student *t* test. Right, The same cells were treated with vehicle or Chloroquine and with increasing concentrations of T-DM1. Cell numbers were estimated by crystal violet staining. Results are averages and standard deviation of five independent experiments. **p* < 0.05, two-tailed Student *t* test; (**C**) Left, Relative HER2 mRNA expression was determined by RT-PCR. Data were normalized to GAPDH mRNA. Averages and standard deviation of four independent determinations are represented. ** *p* < 0.01, *** *p* < 0.001, two-tailed Student’s *t*-test. Middle, Right, Lysates were analyzed by Western blot with the indicated antibodies and densitometrical quantification was performed. Quantitative results are expressed as averages ± standard deviations of three independent experiments; (**D**) Lysates of cells from PDX118, resistant cells or the same cells transduced with a retroviral vector encoding HER2 were analyzed by Western blot with the indicated antibodies and densitometrical quantification was performed. Quantitative results are expressed as averages ± standard deviations of two independent experiments; (**E**) The indicated cell cultures treated with increasing concentrations of T-DM1 for 6 days. Cell numbers were estimated by crystal violet staining. Dashed grey line indicates 50% survival and IC50 is shown next to figure legend. Averages and standard deviation of two independent experiments are represented. * *p* < 0.01, two-tailed Student’s *t*-test; (**F**) The indicated cells were treated with increasing concentrations of S-Me-DM1. Cell numbers were estimated by crystal violet staining. Results are averages and standard deviation of five independent experiments. * *p* < 0.05, ** *p* < 0.01, *** *p* < 0.001, two-tailed Student *t* test; (**G**) The indicated cell cultures were treated with vehicle or Reversan and with increasing concentrations of T-DM1. Cell numbers were estimated by crystal violet staining. Results are averages and standard deviation of five independent experiments. **p* < 0.05, two-tailed Student *t*-test.

**Figure 4 cancers-12-00670-f004:**
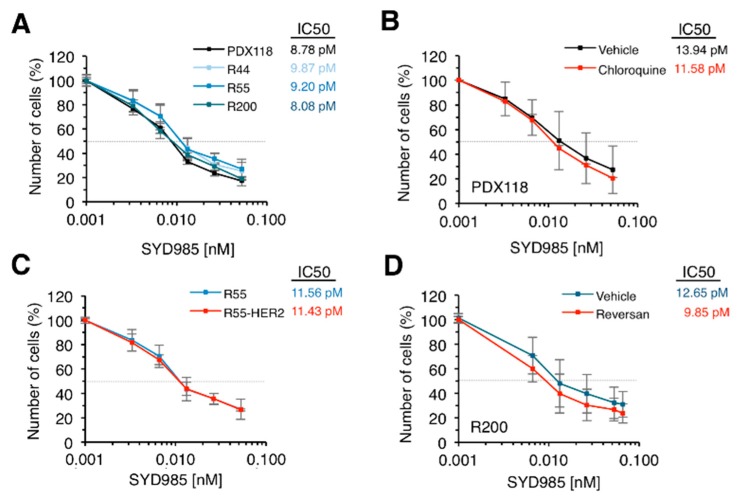
Effect of SYD985 on in vitro resistant models. (**A**) The indicated cell cultures were treated with increasing concentrations of SYD985 for 6 days. Cell numbers were estimated by crystal violet staining. Dashed grey line indicates 50% survival and IC_50_ is shown next to figure legend. Averages and standard deviation of six independent experiments are represented; (**B**) Cultures from PDX118 were treated with vehicle or chloroquine and with increasing concentrations of SYD985. Cell numbers were analyzed as in A; (**C**) The indicated cells were treated and analyzed as in A; (**D**) R200 cells were treated with vehicle or Reversan and with increasing concentrations of SYD985. Cell numbers were analyzed as in A.

**Figure 5 cancers-12-00670-f005:**
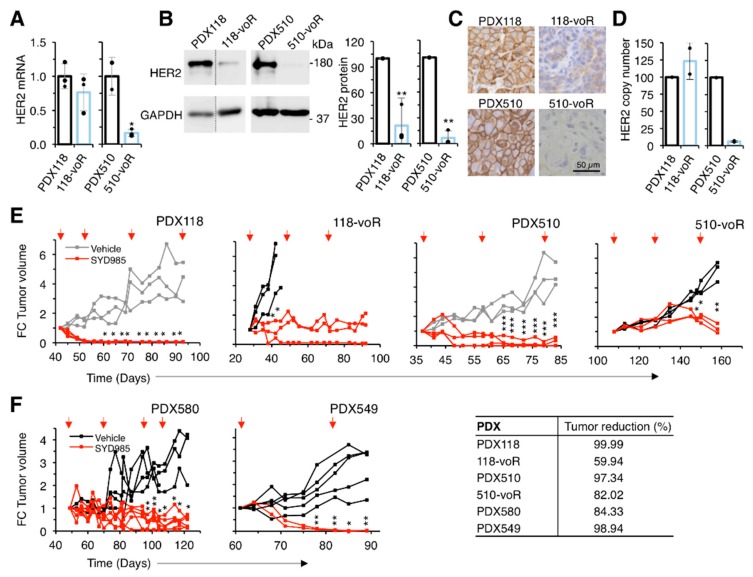
Characterization of in vivo resistant models and effect of SYD985. (**A**) Relative HER2 mRNA expression was determined by qRT-PCR. Data were normalized to GAPDH mRNA. Averages and standard deviation of three independent determinations are represented; (**B**) Lysates were analyzed by Western blot with the indicated antibodies (left panel) and densitometrical quantification was performed (right panel). Quantitative results are expressed as averages ± standard deviations of three independent experiments. ** *p* < 0.01, two-tailed Student’s *t*-test; (**C**) Immunohistochemical analysis of the expression of HER2; (**D**) HER2 gene copy number determination by qPCR; (**E**, **F**), NOD/SCID mice carrying the indicated PDXs were treated with vehicle or SYD985 as indicated. Tumor volume was determined and expressed as fold-change relative to the initiation of treatment. Note that the volumes corresponding to PDX118 and PDX510, depicted in gray, treated with vehicle are the same as those shown in Figure 1D. In order to minimize the number of mice used, treatments with T-DM1 and SYD985 were done in the same experiment, although they are presented in different figures. Tumor reduction was calculated at the end of the experiment as the percentage of the average tumor volumes of the tumors treated with SYD985 relative to tumors treated with vehicle. * *p* < 0.05, ** *p* < 0.01, *** *p* < 0.001 two-tailed Student’s *t*-test.

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
