# Peer review of "The Second Generation Antibody-Drug Conjugate SYD985 Overcomes Resistances to T-DM1"

_cancers, 2020, doi:10.3390/cancers12030670_

Round 1

Reviewer 1 Report

The authors tested the effect of  ADCs, 25 SYD985 on tumors resistant to T-DM1 by developing several patient-derived models of resistance to the drug. These models including in vivo, in vitro, mRNA expression,cell proliferation assay, protein production, etc. 

In my opinion, the study is very interesting and deserves to be published. 

minor issue: for Paragraph

"It should be kept in mind that this observation may represent an underestimation
239 of the frequency of HER2 downmodulation in resistant tumors, as our results show that HER2 can
240 be downmodulated not only by loss of gene amplification but also through post-transcriptional
241 mechanisms in cells that have not lost gene amplification."

Line 192 does not detail te downmodulated through the post-transcriptional mechanism.

Minot issue: By loss of gene ==> by the loss of gene

Author Response

Reviewer 1

We sincerely thank the reviewer for evaluating our manuscript.

The authors tested the effect of ADCs, SYD985 on tumors resistant to T-DM1 by developing several patient-derived models of resistance to the drug. These models including in vivo, in vitro, mRNA expression, cell proliferation assay, protein production, etc. 

In my opinion, the study is very interesting and deserves to be published. 

minor issue: for Paragraph

"It should be kept in mind that this observation may represent an underestimation
239 of the frequency of HER2 downmodulation in resistant tumors, as our results show that HER2 can
240 be downmodulated not only by loss of gene amplification but also through post-transcriptional
241 mechanisms in cells that have not lost gene amplification."

Line 192 does not detail te downmodulated through the post-transcriptional mechanism.

The analysis of HER2 levels by gene amplification can lead to an underestimation of the frequency of HER2 downmodulation. This is reflected in one of our resistant models (118-voR), which presents decreased HER2 protein levels while mRNA levels and gene copy number remain unaffected. Several mechanisms regulating RNA levels and stability could be involved in this downregulation and, later on, protein levels can also be regulated by post-translational modifications and protein degradation. Unfortunately, elucidation of these mechanisms is out of the scope of this manuscript.

Minot issue: By loss of gene ==> by the loss of gene

We have modified this sentence in the lines 246 and 248 of the latest version of the manuscript.

“…that 30% displayed reduced HER2 expression through the loss of gene amplification (28). It should be kept in mind that this observation may represent an underestimation of the frequency of HER2 downmodulation in resistant tumors, as our results show that HER2 can be downmodulated not only by the loss of gene…”

Reviewer 2 Report

The publication titled "The second generation antibody-drug conjugate SYD985 overcomes resistances to T-DM1" is an interesting and innovative experimental work. It has a scientific aspect, and in the future, after expanding research, it can also have a practical aspect and be used in cancer therapy.
The graphic documentation of the results (figures 1-5) should be enlarged, because in this form the figures are hardly legible. The legend of the figures is very detailed. This is a big advantage of this publication.
The negative side of the developed publication is a very laconic description in subsection 4.2. How many breast cancer patients were used in the developed results? How many biopsies were taken and how many tumor samples? How many Authors did the mice use?
In the references chapter, current publications from the most of the last 5-10 years are given.

Author Response

We sincerely thank the reviewer for evaluating our manuscript.

The publication titled "The second generation antibody-drug conjugate SYD985 overcomes resistances to T-DM1" is an interesting and innovative experimental work. It has a scientific aspect, and in the future, after expanding research, it can also have a practical aspect and be used in cancer therapy. The graphic documentation of the results (figures 1-5) should be enlarged, because in this form the figures are hardly legible. The legend of the figures is very detailed. This is a big advantage of this publication.

The figures have been enlarged and in figure 3 and figure 5 the panels have been moved to make them bigger. The negative side of the developed publication is a very laconic description in subsection 4.2. How many breast cancer patients were used in the developed results? How many biopsies were taken and how many tumor samples? How many Authors did the mice use?

The patient-derived xenografts (PDXs) showed in this manuscript were derived from 5 breast cancer patients. PDX515 and PDX549 were originated from a biopsy of the primary tumor and PDX118, PDX510 and PDX580 were generated from a biopsy of skin, a biopsy below the reconstruction and a lymph node biopsy, respectively. One to two mice were used per sample to implant the patient biopsies, depending on its size. This information has been added to subsection 4.2 and subsection 4.3 to clarify the description as detailed below:

“4.2. Tumor samples

Human breast tumors used in this study were from biopsies or surgical resections at Vall d'Hebron University Hospital and were obtained following institutional guidelines. PDX515 and PDX549 were originated from a biopsy of the primary tumor and PDX118, PDX510 and PDX580 were generated from a biopsy of skin, a biopsy below the reconstruction and a lymph node biopsy respectively. The IRBs at Vall d’Hebron University Hospital provided approval for this study in accordance with the Declaration of Helsinki. Written informed consent was obtained from all patients who provided tissue

4.3. Patient-derived xenografts (PDXs)

Six to eight-week old NOD.CB17-Prkdcscid/J (NOD/SCID) mice were purchased from Janvier Labs. Mice were maintained and treated in accordance with protocols approved by the Ethical Committee for the Use of Experimental Animals at the Vall d’Hebron Institute of Oncology. Fragments of human breast tumors were implanted into the fourth fat pad of the mice, which were supplemented with 1 μM 17ß-estradiol (Sigma-Aldrich) in the drinking water. One to two mice were used per sample to implant the patient biopsies, depending on its size. “

In the references chapter, current publications from the most of the last 5-10 years are given.

We have chosen the references that have been considered most relevant to the purpose of this study and selected the original articles that show what is referenced, independently of the date of publication.